# Assessment of Disused Public Buildings: Strategies and Tools for Reuse of Healthcare Structures

Lorenzo Diana [1,*], Saverio D'Auria [1], Giovanna Acampa [2,3] and Giorgia Marino [2]

1 DICEA—Department of Civil, Building and Environmental Engineering, University of Naples Federico II, 80125 Naples, Italy; saverio.dauria@unina.it

2 Faculty of Engineering and Architecture, University of Enna Kore, Cittadella Universitaria, 94100 Enna, Italy; giovanna.acampa@unikore.it (G.A.); giorgia.marino001@unikorestudent.it (G.M.)

3 DIDA—Department of Architecture, University of Florence, 50100 Florence, Italy

* Correspondence: lorenzo.diana@unina.it; Tel.: +39-081-7682-144

**Abstract:** The aim of this paper, in line with the 2030 European Agenda and 2021 Italian "Recovery and Resilience Plan" objectives, is to define an evaluation methodology and tool to support public administrations to detect buildings, currently unused or underused, that might be apt for transformation interventions. The focus is on historical Italian healthcare buildings since these show widespread decay and neglect. A five-step methodology has been developed: screening of public assets; classification, evaluation, and identification of buildings, based on the "potential index"; GIS mapping and inventory; selection of buildings for in-depth analysis; BIM digitization and definition of the "transformability index". In the fifth phase, an evaluation tool is integrated into the BIM software to automatically calculate the transformability index of each building using six indicators: usability, fragmentation, modifiability, roof implementation, external envelope, and window-to-wall ratio. The "transformability index" helps define the most appropriate buildings to intervene with for reuse. Building transformation is connected to construction features and layout organization and it is limited by architectural, structural, and artistic constraints.

**Keywords:** disused buildings; reuse; assessment tools; evaluation; transformability; BIM; healthcare structures; typology; layout reconfiguration





## 1. Introduction

Italian public real estate is made up of a very large number of buildings, over one million units, of which more than two-thirds are residential and the remainder institutional. According to the Italian Ministry of Economy and Finance [1], their economic value is around 300 billion euros, but approximately 10% of them are disused and only half of those used are a source of revenue. Furthermore, in Italy, around 60% of the buildings were constructed before 1970 [2], prior to the introduction of laws on energy consumption control in buildings [3] and construction in seismic regions [4]. It thus becomes clear how complex the management of this large estate is in terms of ordinary maintenance or even forecasting energy, functional, and structural renovations.

Focusing on the Campania region, the public real estate amounts to about 5000 non-residential or commercial units (with the exclusion of homes, garages, warehouses, shops, industrial and craft buildings, land, and archaeological sites). Almost 80% of them were built before 1980 and nowadays are vacant or in very bad conditions. Many of them (around 1900 [5]) are listed due to their recognized architectural and cultural values.

Hospitals and healthcare structures [1] account for a significant share of the overall public real estate and show widespread structural and technological decay. Since 2020, the global COVID-19 pandemic has added a further issue to this complex scenario, highlighting how the weak local healthcare facility network has hampered the management of the crisis. As a secondary effect of that, it has been estimated that in the first five months of 2020, about

1.4 million healthcare screening sessions were not carried out, thus seriously reducing the detection of early-stage cancer pathologies [6].

In this context, structural and technological improvements to and functional reuse of public buildings are badly needed, all the more in light of the provisions of the Italian "Recovery and Resilience Plan", financed by European funds for the Next Generation EU and approved by the Council of Ministers on 12 January 2021 [7]. Public administrations are called on to develop building and urban regeneration projects. The use of DSSs (decision support systems), such as multi-criteria methods, is considered in the "guidelines for defining the technical and economic feasibility of projects for public work contracts", of the Recovery and Resilience Plan and the National Plan for Complementary Investments [8], as a prerequisite for projects to get the go-ahead. Hence, appraisal and its tools are in increasing need [9]. This paper aims to define an evaluation methodology and tool to support the management, monitoring, and determination of intervention priorities for public administrations.

To this end, a list of indicators has been developed, starting at the large scale and even including building layout configuration, which is useful for evaluating the propensity of public buildings to be transformed and reused. Automatic indicator calculations are then introduced to the BIM environment, integrating the evaluation tool into the software REVIT. This procedure is applied to heritage healthcare structures, in particular, the monumental hospitals of the city of Naples (Campania).

### 1.1. State-of-the-Art Public Investments

Urban regeneration is a priority in sustainable and inclusive policies. It involves architectural, environmental, energy, and social redevelopment processes (starting from the reuse of existing real estate assets) that may have important social and economic impacts such as the transformation of degraded urban areas into creative and innovative incubators.

The regeneration and efficient management of disused buildings can significantly influence the "urban context in its location and possible historical and artistic value, thus constituting a precious resource, not only in immediate monetary terms, but also as an element for requalification and growth of large portions of cities [ . . . ]" [10]. Furthermore, this might help to improve the quality of the social and healthcare services offered to the population [11] and citizens' social and psychological conditions [12]. Even the UN, through the 17 Sustainable Development Goals (SDGs), defining a strategy "to achieve a better and more sustainable world for everyone", highlighted in goal number 11 "Cities and Sustainable Communities" the need to "enhance inclusive and sustainable urbanization and capacity for participatory, integrated and sustainable human settlement planning and management in all countries", "[ . . . ] strengthen efforts to protect and safeguard the world's cultural and natural heritage ", and "[ . . . ] increase the number of cities and human settlements adopting and implementing integrated policies and plans towards inclusion and resource efficiency [ . . . ]" [13].

In this spirit, the Italian "Recovery and Resilience Plan" allocates significant resources to urban regeneration actions both in small- and medium-sized cities and in metropolitan areas [14]. Two out of the plan's six missions are aimed at "Inclusion and Cohesion" (for which 27.63 billion euros are allocated), which includes efforts to reduce marginalization and social degradation by redeveloping public areas, and "Health" (19.72 billion euros), which has the purpose of improving hospital infrastructures, thus improving the local assistance offered throughout the country.

### 1.2. Strategies for Sustainable Regeneration Intervention: State-of-the-Art Heritage Buildings

The need for extensive regeneration action on public real estate—in use, disused, or even abandoned—can be met through significant work to classify and assess buildings based on appropriate categories and indicators.

Concerning the existing modes of building quality assessment, building performance evaluation (BPE) provides a solution [15]. It establishes appropriate assessment procedures

for each phase of the building's life cycle to be carried out in a loop: planning (phase 1); programming (phase 2); design (phase 3); construction (phase 4); occupancy (phase 5); reuse or recycling (phase 6). This paper deals with phases five and six (occupancy and reuse/recycling).

Several tools have been developed in the last decades to support decision-makers to assess the effectiveness and quality of retrofitting interventions, especially in the field of multicriteria evaluation methods [16]. Concerning design quality, several research studies have been carried out in Europe [17,18], especially in Italy, with the Siva-SISCo evaluation method [19,20], while EPiQR [21,22], developed at the EPFL, has been found particularly fitting to evaluate the environmental impact of retrofitting existing buildings.

Concerning historical heritage, the first tool to introduce the inventorying of public heritage buildings was InterSAVE [23,24]. Taking the cultural landscape approach, it considers heritage buildings as part of a quality urban system and continuous structure and classifies them with photos and a short description. Nowadays, topics related to heritage preservation are often connected to energy-consumption savings, environmental impact, and seismic risk reduction. The approach introduced in InterSAVE should, therefore, be adjusted based on such strategies.

Data collection, surveys, and assessment of public heritage buildings are instrumental to management plans, especially when dealing with partially or totally abandoned structures. Valuing cultural heritage has always been a key research topic, focusing mainly on economic aspects [25–28] and the overall conservation state [29]. In Italy, following the approval in 2004 of the "code of cultural heritage and landscape" [30], valuation and preservation strategies are held as equally significant, although valuation, even for economic exploitation, has to be functionally compatible with the architectural and structural layout. In 2019, Konsta [31] highlighted how functional compatibility is the key aspect underpinning the success of heritage building reuse interventions. Conejos called in 2013 [32] for the analysis of the threats and opportunities provided by the building, taking into consideration the surrounding environment and the community's expectations of a building's function.

Reuse of abandoned buildings proves to be successful when involving stakeholders and surrounding communities, who may benefit in terms of new jobs, better social identity, and fewer crimes [33]. This should be preferable to demolition/reconstruction [34] not only for heritage buildings—that have been already singled out for preservation [35]—but also for ones that are not listed, to recycle the original environmental resources employed [36]. Reuse initiatives extend the operational lives of buildings vastly, reducing the amounts of waste materials that are unavoidably produced when demolishing/reconstructing [37]. It is the first step toward a circular economy where waste is minimized [38] since using recycled products in the construction industry still meets several barriers [39].

For heritage buildings, Wang and Zengo in 2010 [40] listed six criteria for prioritizing the functions to consider (cultural, environmental, economic, social, architectural, continuity). This approach tends to rely on the ARP (adaptive reuse potential) model proposed by Langstone et al. in 2008 [41], which evaluated the potential reuse of buildings by looking at their observed obsolescence (physical, economic, functional, technological, social, legal) and thus estimating their future useful life. In 2011, Conejos et al. [42] proposed a tool to rank the reuse of listed buildings (adaptSTAR), referring to a long list of criteria grouped into seven categories (physical, economic, functional, technological, social, legal, political). It must be stressed that in multicriteria reuse approaches, certain key factors can describe buildings in a basic but quick approach [43,44], such as the degradation level assessment of the envelope façade proposed by Rodrigues et al. in 2019 [45]. In Table 1, a short summary of the literature comparison concerning evaluation criteria for heritage building reuse is depicted. For each criterion, several qualitative or quantitative indicators have been identified. Further detail can be retrieved from the references reported in Table 1.

**Table 1.** Criteria for heritage building reuse retrieved from the analyzed literature.

| Cultural | Environmental | Economic | Market | Social | Physical | Functional |
|----------|---------------|----------|--------|--------|----------|------------|
| [40,43] | [40,43] | [40–43] | [43] | [40–42] | [41,42] | [41,42] |
| Continuity | Architectural | Changeability | Legal | Technological | Political | Location |
| [40] | [40] | [43] | [41–43] | [41,42] | [42] | [43] |

As for energy consumption reduction, the new regulations often conflict with the existing building and power systems, requiring works that alter essential architectural elements [46]. On this issue, the 2015 "Italian guideline for energy efficiency enhancement of cultural heritage" [47] provides useful guiding principles [48] for listed buildings, where such works should be carried out with techniques compatible with the building's construction and artistic and cultural characteristics [49,50]. Such approaches should be adopted also for seismic risk mitigation [51] and fire regulation [52].

The above literature review shows that some intangible aspects (economic, legal, and social) seem to be critical to reuse interventions, and the trend is to face this issue from a managerial perspective. Yet, aspects related to the physical and technological feasibility of interventions are less dealt with. This gap is one of the main reasons behind this paper, which focuses on the continuity and ease of adaptation of buildings to new functions, taking into consideration the space layout, condition of services and systems, functional changeability, and technological constraints.

*1.3. Support for BIM in the Regeneration of Existing Buildings*

BIM (building information modeling) is a model that contains all the information concerning the entire life cycle of a construction, from project planning to building and use, until demolition [53,54]. In the construction sector, BIM is increasingly used to facilitate interoperability and facility management among project partners [55,56]. Although a BIM model can interact with different disciplines, presently, it is mostly applied to new construction rather than to existing buildings. On the other hand, HBIM (heritage building information modeling), even if there may be problems with translating graphics into data [57], can offer benefits for the management of the historic built environment by storing information about building components and their classification and then generating automatic results [58].

In general, the HBIM implementation process is based on the correct graphic representation of architectural components to be identified as parametric objects of the model. They are mapped through advanced 3D survey techniques, such as digital photogrammetry and laser scanning, thus improving the digitization process of built heritage [56]. According to Murphy [59], this process includes several steps to get to the final model, from identifying heritage details to defining data collection procedures, modeling concepts in 3D, building parametric historic components/objects, and finally, carrying out as-built BIM, where these objects are mapped onto scan data, producing the final product [60,61].

However, the greatest difficulty to overcome is finding a real standardization process, as historic buildings may have various layers, additions, demolitions, changes of use/function, etc. [57,62].

Interesting studies have focused on improvements to the HBIM methodology. Saygi et al. [63] evaluated the impact of BIM and GIS, highlighting their advantages and disadvantages for managing heterogeneous data to create a model of the historical buildings of the Kurşunlu Khan in Turkey. Murphy et al. [64], in their first experiments, used a new approach for semi-automatic modeling of building façades from laser or image data using a customized plug-in for the ArchiCAD BIM software; later, they attempted to improve the integration of information into HBIM, uploading the finished BIM model to CityGML (external international GIS platform), thus facilitating management and further analysis [59]. Baik's JHBIM methodology focused on creating the Hijazi architectural elements library

based on laser scanner and image survey data, using Autodesk Revit as the main BIM platform. This included information about the history, degradation, or materials, which combined to create an Old Jeddah-specific object library, to be used for any heritage project in Al-Balad district, Jeddah City [65]. Nagy et al. argued that the usage of smart sensing would change the way heritage buildings are managed and preserved by visualizing the energy performance through the HBIM platform [66]. It appears straightforward that planned preventive actions can minimize the progress of degradation and the cost of future interventions [67], thus extending monuments' lifetimes and preserving built cultural heritage assets for future generations [68].

To summarize, BIM is generally applied to new buildings, while for existing historical ones, HBIM seems the most appropriate tool, though a clear standardization process is difficult to define. In the scope of this paper, a current research gap is addressed, which is the application of BIM as a guide to detect intervention strategies and priorities.

## 2. Methodology

The methodology (Figure 1) consists of five macro-phases: (1) screening public assets; (2) classifying, rapidly evaluating, and identifying buildings based on the "potential index"; (3) GIS mapping and inventory; (4) selecting buildings for in-depth analysis; (5) BIM digitization and defining the "transformability index".

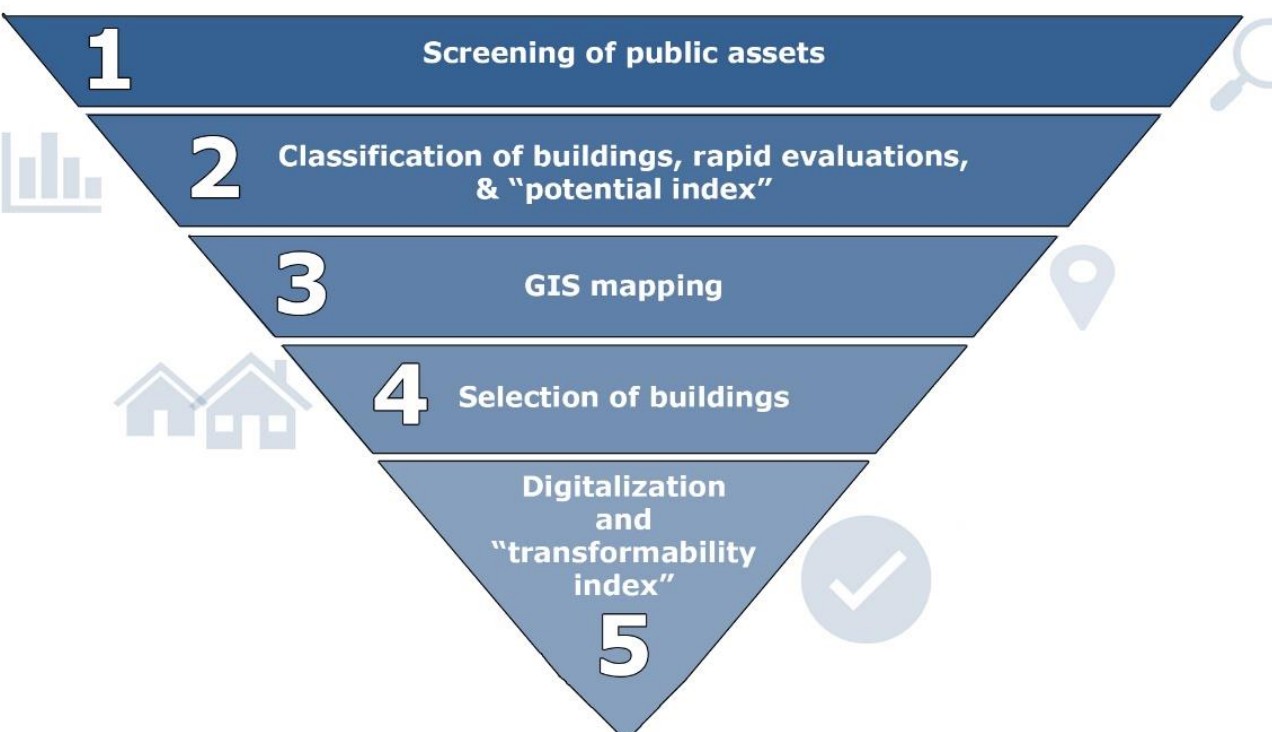

**Figure 1.** Workflow of the proposed methodology.

Phase 1: collecting public building information (geographical location, cadastral identification, local ownership, construction period, functional use, gross area, state of conservation, etc.) in a database. Relating to Italy, the databases of public real estate available on the Ministry of Economy and Finance website are appropriately integrated with data from local administration databases (regions, provinces, municipalities, local healthcare agencies, etc.).

Phase 2: selecting unused or underused buildings (at least larger than 500 square meters, as the focus is on possible future compatible uses). Further detailed information is retrieved through rapid visual inspections based on a survey form specifically developed for this research (Appendix A). For each building, a "potential index" is then defined by

processing data and judgments collected in the above form and appropriately transformed into numerical evaluations. This index allows for hierarchical classification of the buildings liable to regeneration, underling their potential for future reuse. The "potential index" is described in-depth in Section 2.1. All data and judgments of this phase are stored in the global database.

Phase 3: uploading the database information to the GIS environment via IDW (inverse distance weighting) interpolation, thus creating a colored map where the sample points (i.e., the identified buildings) influence the color of the map (from red to green) according to the "potential index". Areas where the buildings have a low "potential index" are colored tending toward red while areas in which buildings have a high "potential index" are colored tending toward green.

Phase 4: identifying the properties that may raise the interest of the public administration when subsequently selecting buildings for in-depth analyses aimed at transforming them. The selection is based on the score of the "potential index", the public administration that controls the geographic location involved, and on processing other collected data.

Phase 5: modeling the selected buildings in the BIM software, allowing for an in-depth check of their transformability and final validation of the buildings to be regenerated. By adopting a few key indicators, a "transformability index" may be defined, to hierarchically classify which buildings are apt for regeneration. Further studies, based on parametric model checking processes that refer to the technological-functional requirements of the future intended use, will be the object of future research.

*2.1. Focus on Phase 2: Potential Index*

The data collected cover the:

1. State of conservation and maintenance, focusing on the following elements: floors; walls and ceilings; fixtures; electrical system; water system and sanitation; heating system; common areas (accesses, stairs, lifts, façades, roofs, and common parts in general). "The state of conservation of the building is considered mediocre if, out of the above-mentioned elements or groups of elements, three are in poor condition [ ... ], poor if four of them are in poor condition [ ... ] or if the real estate unit does not have an electrical system or water system with running water in the kitchen and services, or if it does not have private toilets [ ... ]" [69] (art. 21of Italian Law 392/78);

2. Context; the score is related to the geographical location and depends on the environmental quality, availability of public services, general perception of safety, etc. If the building is located in an industrial area the score is 1, in an agricultural area 2, 3 in the historic center, 4 in a commercial area, and 5 in a residential area;

3. Accessibility; the score is related to the difficulty of reaching the building both by public and private transportation. For public transport (which is weighted 70% compared to 30% for private transport), the distances from the bus, metro, and railway stations are considered, and for private vehicles, the types of roads and the presence of nearby parking lots. Scores are determined as a function of the distance $r$ (see Table 2, top left);

4. Availability of services: the score is given according to the proximity to basic services (education, culture, health, commercial areas, etc.) and as a function of the distance $r$ (see Table 2, below);

5. Percentage of the building currently not in use or the "non-use ratio", whereby the less the building is unused, the higher the score. This normalization underlines the convenience of transforming an unused building compared to one partially in use. Scores are defined as a function of the disused area (DA) compared to the total gross area of the building (see Table 2, top right).

**Table 2.** Score normalization for several indicators related to the definition of the potential index.

| Accessibility Score Normalization | | | | Non-Use Ratio Score Normalization | |
|---|---|---|---|---|---|
| Public Transportation Stations | | Parking Lots | | | |
| 1 | $r > 700$ m | 1 | $r > 500$ m | 1 | DA < 30% |
| 2 | $500$ m $< r \leq 700$ m | 2 | $300$ m $< r \leq 500$ m | 2 | $30\% \leq$ DA $< 50\%$ |
| 3 | $400$ m $< r \leq 500$ m | 3 | $200$ m $< r \leq 300$ m | 3 | $50\% \leq$ DA $< 80\%$ |
| 4 | $300$ m $< r \leq 400$ m | 4 | $150$ m $< r \leq 200$ m | 4 | $80\% \leq$ DA $< 100\%$ |
| 5 | $r \leq 300$ m | 5 | $r \leq 150$ m | 5 | DA $\geq 100\%$ |
| **Availability of Services Score Normalization** | | | | | |
| Primary Education | | | Secondary Education | | |
| 1 | $r > 1000$ m | | 1 | | $r > 1500$ m |
| 2 | $700$ m $< r \leq 1000$ m | | 2 | | $1000$ m $< r \leq 1500$ m |
| 3 | $500$ m $< r \leq 700$ m | | 3 | | $850$ m $< r \leq 1000$ m |
| 4 | $400$ m $< r \leq 500$ m | | 4 | | $700$ m $< r \leq 850$ m |
| 5 | $r \leq 400$ m | | 5 | | $r \leq 700$ m |
| Health Services | | | Markets | | |
| 1 | $r > 1300$ m | | 1 | | $r > 1000$ m |
| 2 | $1000$ m $< r \leq 1300$ m | | 2 | | $750$ m $< r \leq 1000$ m |
| 3 | $750$ m $< r \leq 1000$ m | | 3 | | $500$ m $< r \leq 750$ m |
| 4 | $500$ m $< r \leq 750$ m | | 4 | | $400$ m $< r \leq 500$ m |
| 5 | $r \leq 500$ m | | 5 | | $r \leq 400$ m |

The combination of the weighted scores of the five indicators provides the overall "potential index".

### 2.2. Focus on Phase 5: Slight Reuse and Transformability Index

The reuse of historical healthcare buildings is an issue that many large European towns, including those in Italy, have to deal with, such as Rome, Florence, Venice, Milan, and Naples. Due to the implementation in recent decades of highly technological care techniques with great structural and infrastructural impacts, such buildings no longer meet the requirements of modern hospitals, and therefore, stand partially or totally closed.

Yet, their "slight reuse" is proposed, defined as "redevelopment and reuse with partial layout reconfiguration in which the intended use changes while maintaining a similar functional context" [5]. The focus is on intermediate healthcare structures, close to citizen needs, such as hospices [70] and "community hospitals". The Italian "Recovery and Resilience Plan" [7] calls for the creation of 753 community hospitals—one per 80,000 inhabitants—by 2026. Community hospitals play an "intermediate" role between at-home and hospital care; they can relieve larger hospitals of simple health services and reduce unnecessary visits to emergency rooms. Community hospitals can offer temporary hospitalization (15 to 20 days), with low care assistance, and in the contemporary pandemic context, they may be turned into spokes of the healthcare network, ensuring better management of the health risk, guaranteeing continuity of care, and monitoring the population's health [6]. The current literature suggests several tools for carrying out quality evaluations of the healthcare facilities that are in use [71], but an evaluation of future functional reconversions is missing.

When considering large historical hospitals' "slight reuse", two Italian cases can be cited: (i) the restoration, redevelopment, and reuse of the monumental complex of Santa Maria del Popolo degli Incurabili in Naples (Figure 2(a1,a2)); (ii) the healthcare restoration and redevelopment of the hospital complex of SS. Giovanni and Paolo in Venice (Figure 2(b1,b2)).

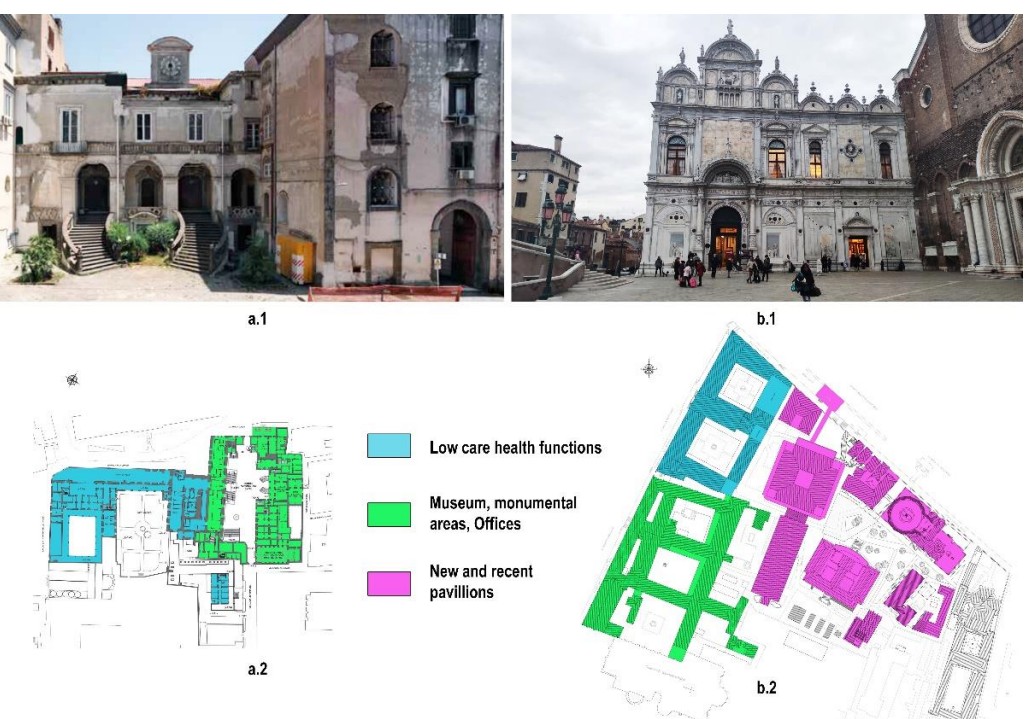

**Figure 2.** Two best-in-practice slight reuse case studies: (**a1**,**a2**) Monumental complex of Santa Maria del Popolo degli Incurabili in Naples (**a1**—photo from the main court taken from [72], **a2**—plan of the new functions); (**b1**,**b2**) Hospital complex of SS. Giovanni and Paolo in Venice (**b1**—photo of the main entrance taken by Lorenzo Diana, **b2**—plan of the new functions).

The Santa Maria del Popolo degli Incurabili complex is the most famous historical hospital in the city of Naples, well known for integrating art, science, health, and charity [73]. To reuse the building, the owner of the hospital (ASL Napoli 1 Centro) intends to set up a low-care "multidisciplinary unit" and create a museum on health by revamping the 18th-century pharmacy. Patient flows—to the "multidisciplinary unit"—and urban and touristic flows—to the museum—may be clearly separated. One of the project's main goals is to enhance open spaces, especially the monumental courtyard, which will play an important role in urban connection.

In SS. Giovanni and Paolo, the focus is on grouping the emergency and intensive departments in recent and new pavilions, thus liberating the two ancient convents from "heavy" functions and turning them into offices, clinics, ambulatories, and intermediate welfare structures. The building's entrance, which is next to the historical medical library of "Scuola Grande di San Marco", will be used for running the library itself, for entry by tourists and local residents [74].

For "slight reuse" projects such as this, the real impacts and transformation of the architectural and construction characteristics of the building should be estimated [75,76] while abiding by the two goals set in the Italian "Code of Cultural Heritage and Landscape", namely preservation and enhancement [30]. To this end, the compatibility with modern building regulations, original plan of the structure, protection of its valuable elements, and relationships and flows that have arisen over the years with the surroundings have to be assessed [77], which requires analyzing a large number of factors.

Applying a method of acknowledged formal analysis may contribute to more rational and transparent intervention processes [31]. Defining the concept of transformability may help. It was defined by Nuti in 2010 [78] as "the intrinsic tendency of a building to modify itself and enable new modes of use; however, this tendency must be realized in a way that is suitable with [ ... ] the typological layout of the pre-existence". This definition is in line with the goal of the present paper, which aims to integrate conservation issues for

listed buildings with the concept of transformability introduced by Acampa et al. [79] for residential buildings.

In phase five, few indicators related to spatial aspects (spatial usability, fragmentation of walls in plan, construction constraints) and construction limitations (implementation of façade and roofs, incidence of windows) define a "transformability index" that measures the compatibility with the original spatial layout and highlights conservation issues.

### 2.2.1. Indicators, Normalization, and Weighting

The transformability index shows decision-makers the propensity to re-use buildings, especially monumental but disused hospitals, while sticking to their function of health assistance. To this end, six specific indicators have been introduced, grouped into two main categories (see Figure 3): plan indicators (usability, fragmentation, constructive modifiability, roof Implementation) and façade indicators (external envelope wall implementation, window-to-wall ratio). Each indicator is analyzed as follows:

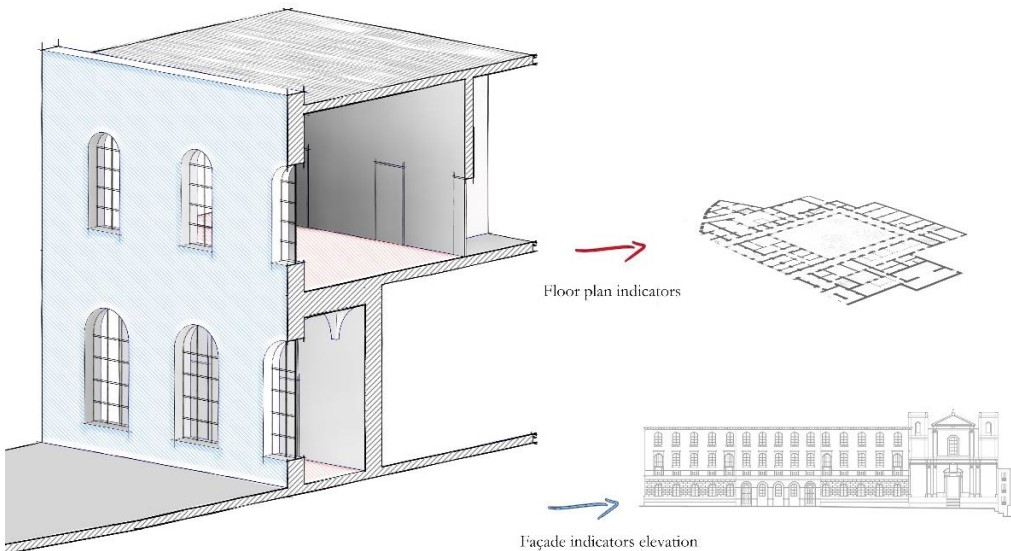

Floor plan indicators

Façade indicators elevation

**Figure 3.** "Gesù e Maria" monumental hospital: isometric view of a portion of the building (**left**); plan and elevation (**right**).

- The usability indicator measures the quantitative relationship between distribution space and served spaces (rooms) [75,80]. Numerically, it is the ratio between the total rooms-served area (excluding corridors/hallways) and net area (rooms + distribution) (1). The surfaces calculation is the net of internal partitions and walls and external spaces. Regarding layout reconfiguration, for healthcare facilities' reuse, on the contrary to housing reconfiguration [79], the lower the usability indicator is, the lower the chances are of transformation. This is because when distribution spaces are large, there is little room space for the required functions.

$$U = \frac{\text{total served areas}}{\text{net area}} [\%] \tag{1}$$

The usability indicator value is normalized to an increasing appreciation curve with concavity pointing upward [19,20]. The higher boundary is at 90% and the lower at 50% assuming that for an area with low usability, connective spaces have a huge impact, meaning less freedom to reorganize the space. In Table 3, single boundaries are shown.

- Next, the fragmentation indicator relates the external and internal borders to the total area available [75], providing information concerning the general layout. Numerically, it is the ratio between the external wall length and internal wall length, with the square

of the gross area (2). Contrary to the formula provided in [79], the denominator is calculated as the square of the gross area, to compare buildings of different sizes.

$$F = \frac{WL_{ext} \times WL_{int}}{\text{gross area}^2} [\%] \quad (2)$$

The fragmentation value is normalized to a decreasing appreciation curve with concavity pointing downward. The higher the internal fragmentation, the lower the opportunities are for transformability and the higher the cost of layout reorganization. Boundary values have been selected looking at other pieces of research where such indicators were applied [79,81], fine-tuning them according to the particular set of heritage masonry buildings here considered. In Table 4, single boundaries are shown.

- The constructive modifiability indicator relates the changeable areas of the plan to the invariant elements [75,80] and provides information on the structural and construction features of the buildings which strongly affect its transformability. It is the ratio between non-modifiable elements' total area (in the plan, identified in structural and plant elements) and gross area (3).

$$CM = \frac{A_{tot,NME}}{\text{gross area}} [\%] \quad (3)$$

Normalization produces a decreasing curve with concavity downward, i.e., transformability decreases as the incidence of non-modifiable elements increases. Reinforced concrete (r.c.) structures—especially r.c. frame structures—have a lower incidence of non-modifiable elements compared to masonry structures, and consequently, higher transformability. On the other hand, in masonry buildings, layout reorganization is limited by the walls and by structural constraints. In this paper, only structural and plant elements have been considered as non-modifiable elements, but in future works, to also evaluate artistic/architectural conservation issues, the calculation may include vaulted ceilings, walls with frescos, and fine floorings. Here, boundary values have been fine-tuned according to the set of heritage masonry buildings considered in Section 3. In Table 5, single boundaries are shown.

- The roof implementation indicator provides information concerning the possibility of installing sustainability devices (such as solar or photovoltaic panels, insulation layers, heat pumps, etc.) on roofs. In the present paper, only the potential for installing solar and photovoltaic panels is considered. For sloped roofs, only non-shaded surfaces and those oriented southward ($\pm 30°$) are considered, while for flat roofs, only non-shaded surfaces (4). The indicator provides information concerning the possible green-energy production after renovation.

$$RI = \frac{A_{tot,NSS}}{\text{roof plan area}} [\%] \quad (4)$$

In this case, normalization produces an ascending proportional line. The value of 50% has been selected as the lower boundary while 100% is the higher one. In Table 6, single boundaries are shown.

- The external envelope wall implementation indicator is one of the two indicators related to façades. It provides information concerning the possibility of intervening with the external envelope walls to add further cladding or insulation panels. It is the ratio between surfaces without any kind of constraint—artistic/architectural or structural—and the whole façade surface (5). The indicator is strictly related to heritage conservation issues and any kind of constraint is considered.

$$EEI = \frac{A_{tot,NCS}}{\text{facade area}} [\%] \quad (5)$$

For the roof implementation indicator, normalization produces an increasing proportional line. The value of 0% has been selected as the lower boundary while 100% is the higher one. In Table 7, single boundaries are shown.

- The window-to-wall ratio, or WWR, provides information on the incidence of the transparent envelope in relation to the total vertical envelope surface [82], thus measuring the impact of the façade structure on energy consumption and internal comfort conditions. A high WWR affects solar gains and daylight in winter and may lead to overheating in summer [83]. In hot regions, optimal WWR values should not exceed 10%, or 20% in a moderate climate and 35–45% in central Europe. In retrofitting listed buildings, due to constraints in adding insulation to the external envelope, often, the only open chance is to operate on the windows. Buildings with a high WWR may provide the opportunity to achieve easy but effective results in energy consumption reduction by changing or restoring ancient single-glazed windows. Previous studies showed [84,85] that the replacement of windows in badly insulated exterior walls (U-value bigger than 1.00 W/m$^2$K) has a significant impact on energy savings for high-WWR buildings. The WWR formula is (6):

$$\text{WWR} = \frac{A_{\text{tot,W}}}{\text{facade area}} [\%] \tag{6}$$

The normalization produces an ascending proportional line. The value of 5% has been selected as the lower boundary while 35% is the higher one. In Table 8, single boundaries are shown.

**Table 3.** Usability score normalization.

| Usability Score Normalization | |
| --- | --- |
| 1 < Score ≤ 2 | 50.0% < U ≤ 72.0% |
| 2 < Score ≤ 3 | 72.0% < U ≤ 80.8% |
| 3 < Score ≤ 4 | 80.8% < U ≤ 86.6% |
| 4 < Score ≤ 5 | 86.4% < U ≤ 90.0% |

**Table 4.** Fragmentation score normalization.

| Fragmentation Score Normalization | |
| --- | --- |
| 5 ≤ Score < 4 | 0.00% ≤ F < 4.40% |
| 4 ≤ Score < 3 | 4.40% ≤ F < 6.16% |
| 3 ≤ Score < 2 | 6.16% ≤ F < 7.28% |
| 2 ≤ Score < 1 | 8.00% ≤ F < 8.00% |

**Table 5.** Constructive modifiability score normalization.

| Constructive Modifiability Score Normalization | |
| --- | --- |
| 5 ≤ Score < 4 | 1.00% ≤ CM < 14.40% |
| 4 ≤ Score < 3 | 14.40% ≤ CM < 19.67% |
| 3 ≤ Score < 2 | 19.67% ≤ CM < 22.85% |
| 2 ≤ Score < 1 | 22.85% ≤ CM < 25.00% |

**Table 6.** Roof implementation score normalization.

| Roof Implementation Score Normalization | |
| --- | --- |
| 1 < Score ≤ 2 | 50.0% < RI ≤ 62.5% |
| 2 < Score ≤ 3 | 62.5% < RI ≤ 75.0% |
| 3 < Score ≤ 4 | 75.0% < RI ≤ 87.5% |
| 4 < Score ≤ 5 | 87.5% < RI ≤ 100.0% |

**Table 7.** External envelope wall implementation score normalization.

| External Envelope Wall Implementation Score Normalization | |
| --- | --- |
| $1 < \text{Score} \leq 2$ | $0.0\% < \text{EEI} \leq 25.0\%$ |
| $2 < \text{Score} \leq 3$ | $25.0\% < \text{EEI} \leq 50.0\%$ |
| $3 < \text{Score} \leq 4$ | $50.0\% < \text{EEI} \leq 75.0\%$ |
| $4 < \text{Score} \leq 5$ | $75.0\% < \text{EEI} \leq 100.0\%$ |

**Table 8.** Window-to-wall ratio score normalization.

| Window-to-Wall Ratio Score Normalization | |
| --- | --- |
| $1 < \text{Score} \leq 2$ | $5.0\% < \text{WWR} \leq 12.5\%$ |
| $2 < \text{Score} \leq 3$ | $12.5\% < \text{WWR} \leq 20.0\%$ |
| $3 < \text{Score} \leq 4$ | $20.0\% < \text{WWR} \leq 27.5\%$ |
| $4 < \text{Score} \leq 5$ | $27.5\% < \text{WWR} \leq 35.0\%$ |

To obtain the transformability index, each indicator is combined with the others through a multi-criteria method. Once all indicators are well-defined in semantic (description of the meaning) and metric terms (through normalization), they should be weighted to assess their relative importance compared to the others. In the present paper, each indicator is assumed to have equivalent importance ($1/6 = 0.166$ each). A final score is thereby assigned, allowing several buildings to be ranked in a hierarchy [86].

### 2.3. Automatic Calculation of BIM Using a Plug-In

To create the building model, the software used here was Revit. The BIM model was obtained starting from 2D drawings and then enriched with the building's geometry, spatial relationships, quantities, and component properties. In our case, the level of definition (LOD) of the model is 200, which is sufficient to evaluate transformability indicators. The 3D model includes interior partitions with their height, length, and area. To automatically calculate the transformability index, a plug-in was developed in the programming language VB.NET combined with Visual STUDIO. The functions for selecting rooms, walls, and windows and for the evaluation process (equations, normalizations, algorithms) were coded, with similar procedures for the indicators to be calculated based on the plan and elevation view. Below are the procedures to be followed for each indicator:

- Usability indicator calculation: select served areas in floor plans and net areas;
- Fragmentation indicator calculation: select both external and internal walls in the plan and gross areas. The values of the lengths of individual external and internal walls are automatically retrieved;
- Constructive modifiability indicator calculation: select both non-modifiable element areas (identified in the plan as structural and plant elements) and gross areas;
- Roof implementation indicator calculation: select areas in which it is possible to install solar and photovoltaic panels, included in the roof plan and total roof area;
- External envelope wall implementation indicator calculation: select (in elevation view) surfaces with and without any kind of constraint (artistic/architectural or structural elements);
- Window-to-wall ratio indicator calculation: select (in elevation view) all windows included in the analyzed façade and walls in which the windows are contained.

Once the selection procedure is complete, calculation is performed automatically. An easy user interface (UI) helps with the procedures and automatically generates a report in .docx, showing the normalized and weighed partial results, and finally, the transformability index.

## 3. Case Study

### 3.1. Campania's Public Stock of Disused Buildings

Campania is the second largest region in Italy for the economic value of public real estate, accounting for almost eight billion euros [87], while the average annual income of Campania families is below the Italian average, at around 25,000 euros per family [88]. This leads to social disadvantage in terms of housing and social welfare. The proposed methodology has been implemented for Campania using public funding (PON AIM project E61G18000550008—activity code AIM1849341-3).

The first screening phase (See Section 2) produced a database of approximately 160,000 units that were geolocated in a GIS environment, showing that they are located mainly around Naples, Salerno, and along the coast (Figure 4a). The data-filtering process produced a set of around 4000 disused or underused buildings suitable for regeneration. Among this set, a sample of 100 buildings was selected to undergo a morphometric, typological, and environmental rapid visual-screening phase (step 2) by filling in the inspection form (see Appendix A) and defining indicator scores. The "potential index" was obtained with AHP [89], weighting the score through expert judgements as follows: (1) State of conservation 0.032; (2) Context 0.178; (3) Accessibility 0.268; (4) Services 0.356; (5) Non-use ratio 0.166. Thus, a dataset containing more specific information for each building was created (see Table 9).

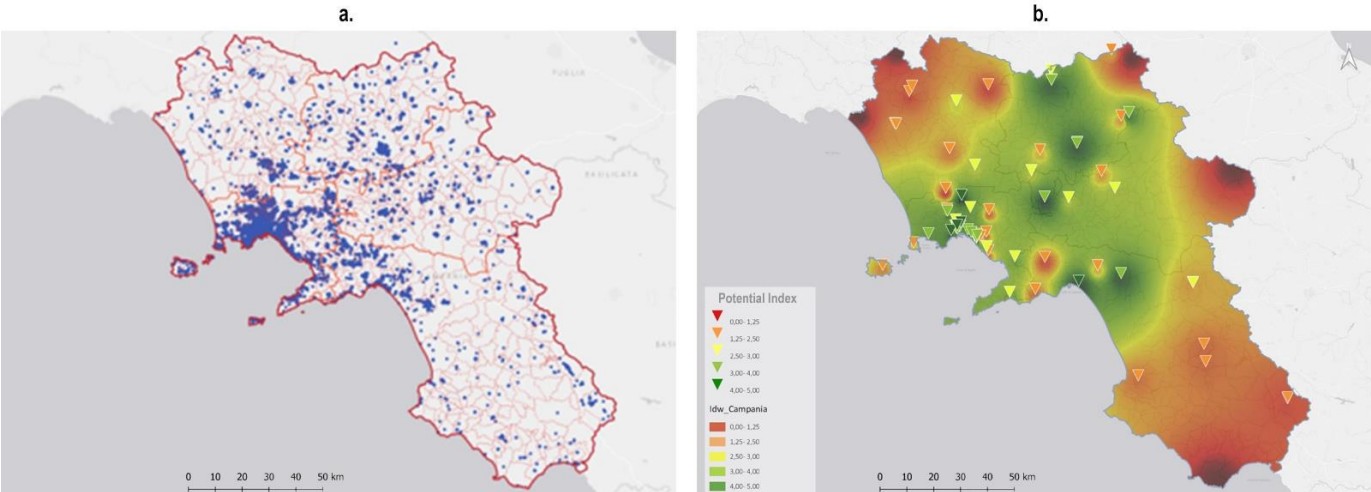

**Figure 4.** (**a**) 160,000 public real estate units in Campania region located in GIS environment; (**b**) chromatic map based on potential index scores.

**Table 9.** Portion of the general database implemented for the Campania region.

| ID | Municipality | Address | Lat. | Long. | ... | Conser. Score | Context Score | Access. Score | Service Score | Non-Use Score | Potential Index |
|---|---|---|---|---|---|---|---|---|---|---|---|
| | | | | | ... | | | | | | |
| 145 | Napoli | Via Santa Maria la Nova 43 | 40.841 | 14.253 | ... | 3 | 3 | 4 | 3 | 2 | **3.07** |
| 146 | Capua | Via Roma 25 | 41.110 | 14.215 | ... | 1 | 4 | 2 | 2 | 4 | **2.14** |
| 147 | Nocera Inferiore | Via Francesco Salimena 120 | 40.748 | 14.640 | ... | 1 | 3 | 2 | 1 | 4 | **1.68** |
| | | | | | ... | | | | | | |

Implementing phase three of the proposed methodology, the GIS environment was developed with IDW interpolation, which produced a qualitative chromatic map (see Figure 4b). Areas with the greatest number of high-potential index buildings are located

near the provinces of Naples, Caserta, and Salerno (dark and light green areas with "potential index" > 3). Given the scope of the present paper and phase four of the methodology, a set of disused and underused healthcare facilities were selected for detailed analysis. In Table 10, this set of buildings is displayed.

**Table 10.** Framework of the analyzed buildings.

| Code-Case Study (Location) | Year of First Construction (Main Renovations) | Original Function | Listed Building | Vertical Structure | Horizontal Structure | Vertical Envelope |
|---|---|---|---|---|---|---|
| SGN—San Gennaro (Naples) | Mid-9th cent. (1468; 1656; end 19th cent., mid-20th cent.) | Monastery/ Hospital | Yes | Masonry walls | Vaults/steel beam+ hollow blocks (or brick vaults) | Tuff |
| ANN— Annunziata (Naples) | 1343 (1433, 1540, 1757, 1888, 1948) | Hospital | Yes | Masonry walls | Vaults/steel beam + hollow blocks (or brick vaults)/wood/hollow blocks + r.c. beams | Tuff |
| GEM—Gesù e Maria (Naples) | 1580 (end 19th cent.) | Convent | Yes | Masonry walls | Steel beam + hollow blocks (or brick vaults) | Tuff |
| FRL—Presidio Frullone (Naples) | 1963 | Office + Healthcare facility | No | R.c. pillars | Hollow blocks + r.c. beams | Double hollow bricks with air space |
| PAL—Ex Biagio Lauro (Palma Campania) | 1965 (1985) | Hospital | No | Masonry walls/r.c. pillars | Hollow blocks + r.c. beams/steel beam + hollow blocks | Tuff/double hollow bricks with air space |
| ALL—Via Allende (Castellamare di Stabia) | 1975 | Healthcare facility | No | R.c. pillars | Hollow blocks + r.c. beams | Double hollow bricks with air space |

### 3.2. Reuse Strategies for the Selected Monumental Healthcare Structures

In phase five of the proposed methodology, 3D surveys and BIM models are created with the goal of computing their "transformability index". For this phase, three monumental Neapolitan hospitals were selected: San Gennaro hospital (SGN) (Figure 5a); Annunziata hospital (ANN) (Figure 5b); Gesù e Maria hospital (GEM) (Figure 5c).

SGN was originally built in the 9th century as the monastery of the nearby church and expanded to become a proper hospital to host plague victims during the 15th and later during the 17th century [90]. It has quite a compact shape: a long four-story double body (corridor/room) encloses three courtyards. The first courtyard—the main one—develops from the south entrance for a considerable length with two sets of 16 arches, today closed with walls or glazed, and two sets of simple windows. The second courtyard's entrance hall is covered by a barrel vault with lunettes full of frescos. The third small courtyard belongs to the church of San Gennaro and includes the façade of the basilica on the north side.

ANN hospital was erected in the 14th century together with the nearby church, convent, orphanage, and housing for single mothers. The building is a complex, compact four-story structure organized around two courtyards. The construction is a stratification of several interventions that took place over the centuries until the end of the 19th century, when with the construction of Corso Umberto, part of the building was demolished [91,92].

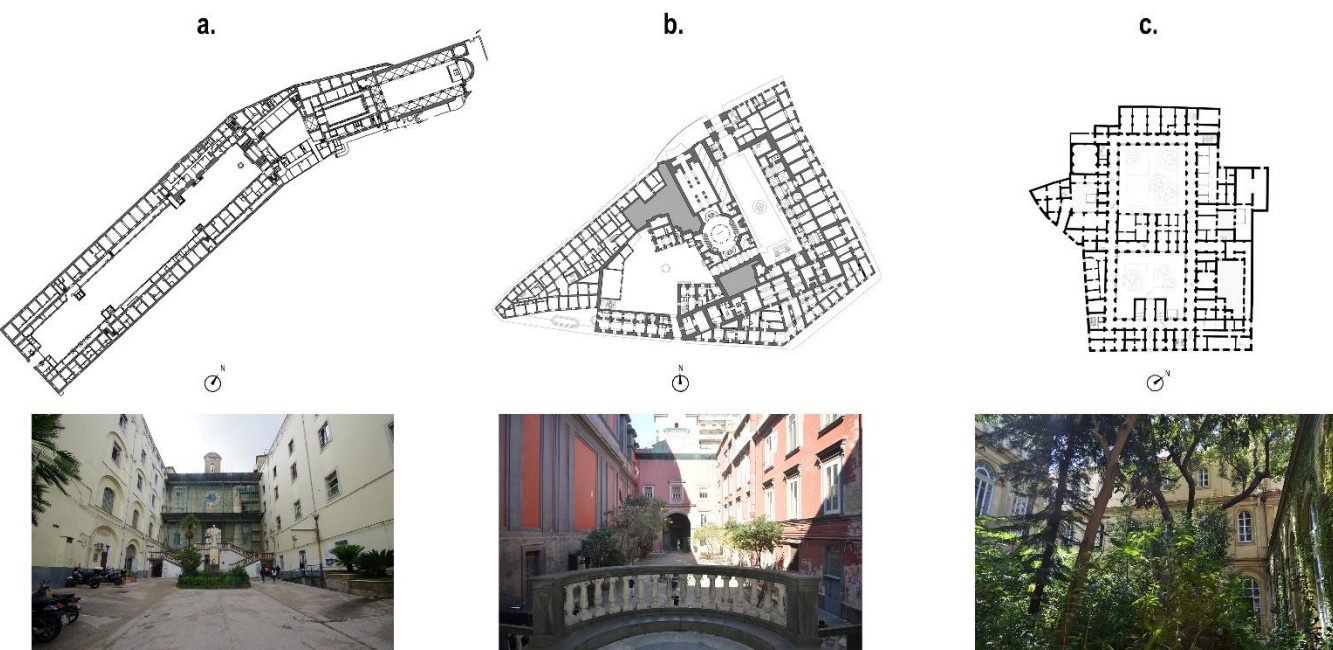

**Figure 5.** Plans and photos of the three case studies: (**a**) San Gennaro Hospital (photo taken by Daniela Volpe); (**b**) Annunziata Hospital (photo taken by Lorenzo Diana); (**c**) Gesù e Maria Hospital (photo taken by Martina D'Alessio).

GEM was originally built as a convent during the 16–17th centuries and heavily transformed into a hospital during the second half of the 19th century. The two/three-story building has a typical convent layout organized around two inner cloisters, a circular portico, and several conventual cells organized on two different floors [93,94]. The 19th-century transformation brought: new entrances on the main façade; new monumental staircases; demolition of the cellars to create big rooms (*camerate*) for patients, especially on the first floor; closure of the original portico running around the cloister to build access corridors to the ambulatories located at the ground floor; separation of the original cloister into two courts by means of a one-story volume in the middle.

### 3.3. Gesù e Maria Monumental Hospital

The plans of the two main floors of the "Gesù and Maria" Monumental Hospital (GEM) are displayed in Figure 6a,b.

The "transformability index" was calculated step-by-step for the whole GEM building. Once the six indicators were defined, the BIM model was generated using Revit, based on in-place visual inspections and drawings retrieved from literature and the archive of the ASL Napoli 1 Technical Office. A portion of the building model is shown in Figure 6e, while in Figure 7a,b, the results for the six indicators of the portion modeled are reported. Then, in Figure 8, the Revit procedure to obtain the score for the U indicator is shown. The six indicators and the transformability index have been calculated for the other two monumental hospitals selected, but the figures are not published here for convenience.

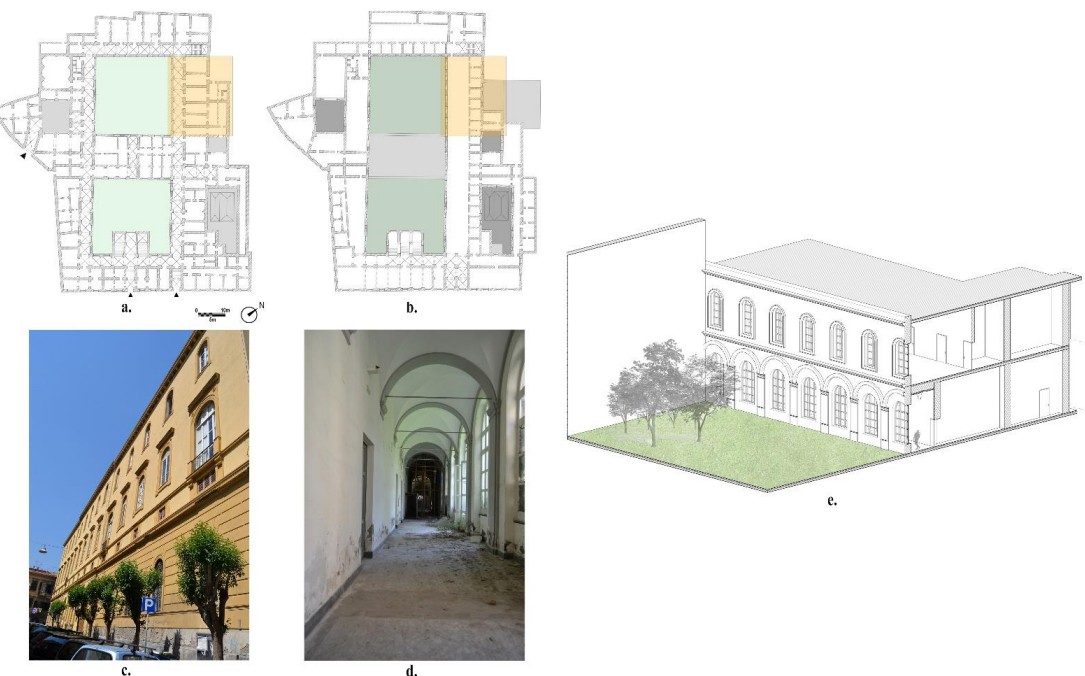

**Figure 6.** Plans, photos, and model of GEM—Gesù e Maria case study: (**a**) ground-floor plan; (**b**) first-floor plan; (**c**) photo of the main façade; (**d**) photo of the closed portico (photos taken by Martina D'Alessio); (**e**) portion of the building model. Figure 6a,b reports in yellow the portion of the building modeled in Figure 6e.

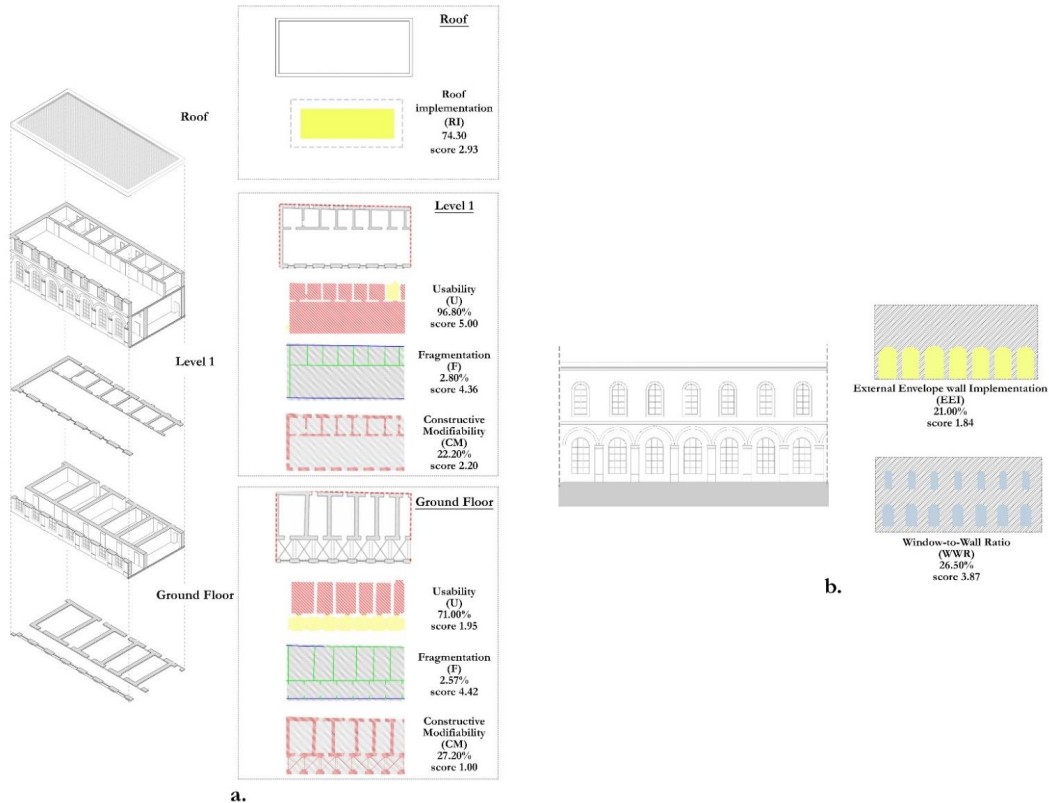

**Figure 7.** (**a**) Results for the plan indicators (U, F, CM, and RI) and (**b**) façade indicators (EEI and WWR) for the portion of the building modeled.

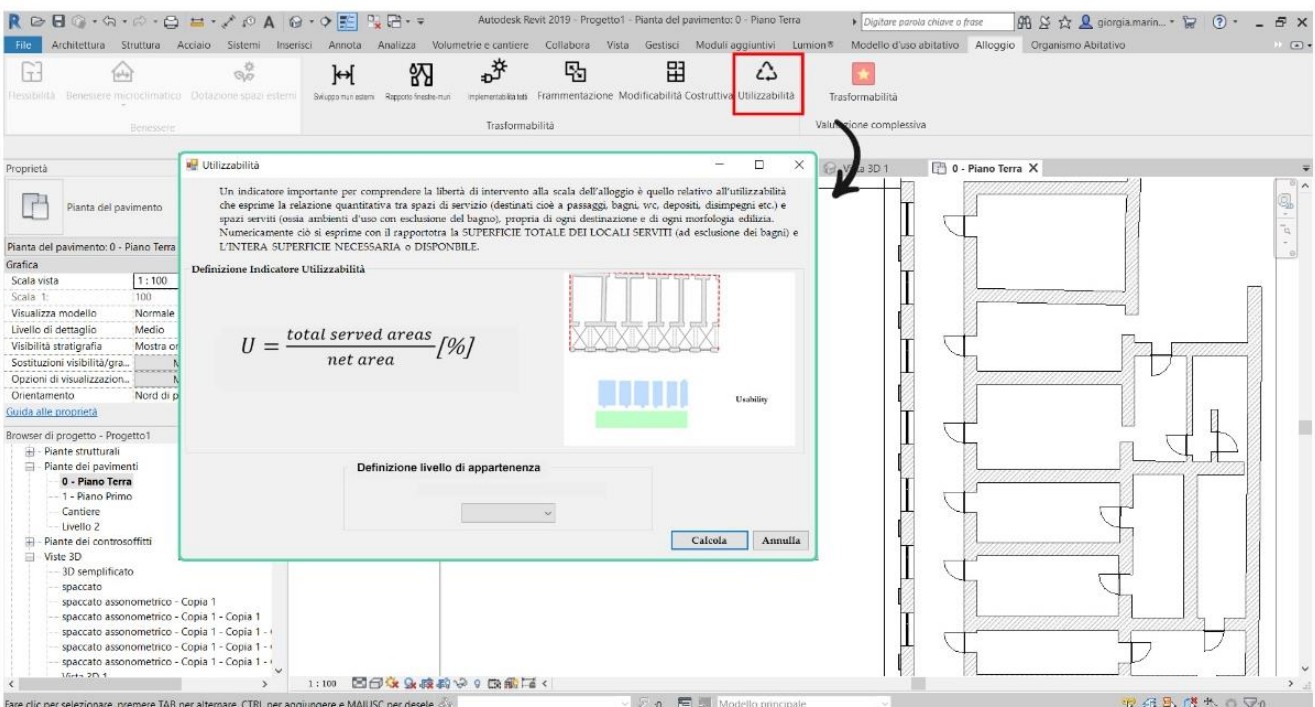

**Figure 8.** Revit plug-in procedure to determine the U indicator score.

## 4. Discussion

### 4.1. Application of the Tool

Applying the methodology to the whole GEM building led to the following results, also shown in Table 11. The low values of the fragmentation indicator (2.69%, F score 4.39) mean that, due to the regular layout distribution, future transformation may take place with a low impact on demolition/reconstruction works. The indicator score is increased by the low fragmentation of the first floor (2.22%, partial F score 4.49), where the 19th-century intervention, carried out to enlarge rooms, demolished many partitions. The masonry structure of the building explains its mid-low constructive modifiability score (2.14). The two floors differ substantially. On the first floor, due to the above-mentioned transformation intervention, the incidence of non-modifiable elements decreases to 19.28% (partial CM score 3.07). If the vaulted ceilings of the corridors running around the cloisters are considered non-modifiable elements (about 900 square meters), the global CM indicator increases to 31.26% (score 1.00). The great incidence of distribution spaces lowers the usability indicator score (1.86), especially on the ground floor where the served spaces weigh on the gross area by 60.22% (partial U score 1.46). Due to the big rooms, this incidence increases to 77.60% for the first floor. For the portion analyzed in Figure 7a, on the first floor, the U indicator reaches 96.80% (score 5.00). The regular shape of the flat roof means there are few shading elements and consequently a high RI score (3.72). Concerning the façade indicators (Figure 7b), the architectural value of the external walls (rustication in the external south façade and a scattered presence of pilasters, cornices, and tympana) lowers the chance of adding external insulation or cladding layers. Therefore, the EEI indicator score is only 1.83. The building has many windows, occupying 19.2% of the vertical envelope (WWR score 2.89). In the north façades, they account for up to around 23%. The six indicators have been weighted equally (0.166), providing a final "transformability index" score of 2.80.

**Table 11.** Indicator scores and "transformability index" for GEM case study.

| | | Portion | | Ground Floor | | Portion | | First Floor | | Global | | |
|---|---|---|---|---|---|---|---|---|---|---|---|---|
| | | Result | Norm. | Result | Norm. | Result | Norm. | Result | Norm. | Result | Norm. | Weigh. |
| Plan indicators | U | 71.00% | 1.95 | 60.22% | 1.46 | 96.80% | 5.00 | 77.60% | 2.64 | 68.95% | 1.86 | 0.166 |
| | F | 2.57% | 4.42 | 3.13% | 4.29 | 2.80% | 4.36 | 2.22% | 4.49 | 2.69% | 4.39 | 0.166 |
| | CM | 27.20% | 1.00 | 25.34% | 1.00 | 22.20% | 2.20 | 19.28% | 3.07 | 22.42% | 2.14 | 0.166 |
| | RI | 74.30% | 2.93 | - | - | - | - | - | - | 84.01% | 3.72 | 0.166 |

| | | Portion | | East Façade | | West Façade | | South Façade | | North Façade | | | | |
|---|---|---|---|---|---|---|---|---|---|---|---|---|---|---|
| | | Result | Nor. | Result | Nor. | Result | Nor. | Result | Nor. | Result | Nor. | | | |
| Façade indicators | EEI | 21.00% | 1.84 | 24.72% | 1.99 | 41.18% | 2.65 | 6.60% | 1.26 | 14.59% | 1.58 | 20.77% | 1.83 | 0.166 |
| | WWR | 26.50% | 3.87 | 19.76% | 2.97 | 16.96% | 2.59 | 19.99% | 3.00 | 22.97% | 3.40 | 19.20% | 2.89 | 0.166 |
| | | | | | | | | | | | **Final Score** | | **2.80** | |

When looking also at the other selected hospitals (see Table 12), GEM has the highest transformability index score (2.80) among the analyzed case studies, with the other two being substantially lower (SGN 2.29 and ANN 2.15). The low value of SGN stems mainly from the layout configuration, construction and structural characteristics, and typological features of the building. The model of the building is based on a double repetition of the corridor + room scheme, which implies a high incidence of distribution spaces and internal partitions. The U, F, and CM indicators drop down to 1.16, 2.49, and 1.03, respectively. In the case of ANN hospital, the low transformability index derives primarily from the external architectural value of the envelope, which lowers the roof and vertical wall implementation (RI and EEI indicators equal to 1.00 and 1.27, respectively). The lack of windows (incidence 9.52% on the whole vertical envelope) impacts the WWR indicator (score 1.60).

**Table 12.** Indicator scores and "transformability index" for Monumental Hospital case study.

| | U | | F | | CM | | RI | | EEI | | WWR | | Transform. |
|---|---|---|---|---|---|---|---|---|---|---|---|---|---|
| | Result | Nor. | Result | Nor. | Result | Nor. | Result | Nor. | Result | Nor. | Result | Nor. | Index |
| SGN | 53.51% | 1.16 | 6.73% | 2.49 | 24.75% | 1.03 | 93.48% | 4.48 | 40.59% | 2.62 | 12.29% | 1.97 | **2.29** |
| ANN | 66.76% | 1.76 | 4.18% | 4.18 | 18.58% | 3.21 | 32.57% | 1.00 | 6.80% | 1.27 | 9.52% | 1.60 | **2.15** |
| GEM | 68.95% | 1.86 | 2.69% | 4.39 | 22.42% | 2.14 | 84.01% | 3.72 | 20.77% | 1.83 | 19.20% | 2.89 | **2.80** |

To check the reliability of the proposed methodology, the transformability index was also calculated for three healthcare facilities, which are partially disused but not listed and without any architecturally or artistically relevant components (see Table 10 for abbreviations). They are r.c. frame buildings dating to the second half of the 20th century, except for PAL, which has a small masonry section.

They have significantly higher transformability indexes (see Table 13). Specifically, FRL and PAL almost reach 3.5, while the ALL building reaches 3.29. Certainly, the punctual vertical load-bearing structure (r.c. pillars) increases the degree of freedom, mainly connected with the CM indicator. In addition, the fact that these case studies are not listed buildings implies higher scores for the EEI indicator. The high values of the transformability index are balanced by lower potential index scores. The FRL, PAL, and ALL buildings are indeed located in peripheral or suburban areas not connected by public transportation or located in areas of low environmental value. For monumental hospitals, SGN presents a lower potential index (equal to 2.36) compared to ANN and GEM. SGN's score is in line with FRL, PAL, and ALL since it is in an external and social complex district of the city of Naples while ANN and GEM are in central areas. In particular, ANN is located close to the historic center, increasing the potential index to 3.4.

**Table 13.** Indicator scores and "transformability index" for GEM case study.

| | Year of Construction | Listed | Potential Index | Transformability Index |
|---|---|---|---|---|
| SGN | Mid-9th cent./1468 | Yes | 2.36 | 2.29 |
| ANN | 1343 | Yes | 3.40 | 2.15 |
| GEM | 1580/End 19th cent. | Yes | 3.22 | 2.80 |
| FRL | 1963 | No | 2.35 | 3.46 |
| PAL | 1965 | No | 1.35 | 3.55 |
| ALL | 1975 | No | 2.32 | 3.29 |

*4.2. Main Findings*

From the results thus obtained, it can be underlined that:

- the transformability index is clearly influenced by the type of load-bearing structure and materials employed. For example, the CM indicator provides trivial results when the masonry and r.c. frame structures are compared. The comparison of scores is useful for large-scale analysis when buildings with different features are considered. If only listed buildings are compared—usually masonry structures—the CM indicator provides useful information concerning the internal layout and propensity to undergo transformation;
- within the set of three buildings analyzed, GEM proved to be the more transformable (score 2.8). This is due mainly to its big open spaces on the first floor, created during a 19th-century transformation. ANN proves to be the less transformable building (score 2.15), especially given the artistic and architectural constraints of its roof and façades;
- when, at an early stage, specific objectives for reuse are defined (e.g., energy transformation or layout reconfiguration), the indicators can be weighed to take a different significance. If the transformation is just in terms of energy retrofitting, the RI, EEI, and WI indicators may weigh 0.75 (0.25 each) and the other three indicators (U, F, and CM) 0.25 (0.083 each), leading to different results. In this case, in GEM the score remains the same (transformability index equal to 2.81), for SGN, it increases up to 2.65, while for ANN, it decreases to 1.72. For layout reconfiguration, indicators may be weighed the other way round: the U, F, and CM indicators may weigh 0.75 (0.25 each) and RI, EEI, and WI 0.25 (0.083 each). Again, for GEM, the score does not change (transformability index equal to 2.80). For SGN, it decreases to 1.92, and for ANN, it increases up to 2.58. The weighing process related to early-stage targets may change the priorities of intervention;
- non-listed buildings are more transformable when compared with listed buildings. This depends on structural characteristics (often listed buildings have a masonry load-bearing structure while non-listed buildings have a r.c. frame) and limitations due to the presence of constraints (artistically/architecturally valuable elements) on external walls that hinder the addition of new insulation or cladding layers. High values of transformability indexes for non-listed buildings are balanced by low scores for potential indexes, due mainly to peripheral locations.

**5. Conclusions**

Due to the global circumstances of the COVID-19 pandemic and given the contents of the Next Generation EU (Section 1), it is worthwhile managing, transforming, and regenerating the huge existing public real estate for better and sustainable use. This is true in Europe and especially in Italy with its large quantity of listed buildings. The Italian Recovery and Resilience Plan (Section 1.1) pays particular attention to cultural heritage energy retrofitting and to reuse of disused buildings with the aim of turning them into intermediate care structures. Several evaluation tools and strategies exist in the literature for the adaptive reuse of cultural heritage structures (Section 1.2)—the BIM (and HBIM) proves to be a useful tool to that end (Section 1.3)—but fail to analyze structural, typological,

and construction features of the buildings, and therefore, offer limited insights into future transformations.

Against such a background, the present paper argues for an integration of evaluations at the building level, to support public administrations with real-estate screening and classification and with planning sustainable transformation interventions that evaluate buildings' potential and propensity for reuse. In Section 2, a five-step methodology was described showing that:

- for all buildings, a "potential index" related to contextual evaluations and specific building features can be defined relying on five indicators relating to the presence of services in the surroundings, accessibility, environmental and architectural quality of the context, building conservation, and size of the disused area;
- based on specific targets defined by public administrations, buildings can be selected for further investigations and digitization processes. For selected buildings, the transformability is quantitatively computed by means of six indicators: the usability, fragmentation, construction modifiability, roof implementation, external envelope wall implementation, and window-to-wall ratio. These indicators allow evaluations of aspects related to historical and cultural values, which is meaningful given that the target is mainly listed buildings. The combination of these six indicators defines a "transformability index" that allows us to compare several buildings;
- particular attention is paid to the evaluation of the propensity of a building to be transformed, introducing an index of transformability. It can be computed automatically owing to a plug-in for Revit developed to allow fast assessments, which are particularly useful when many buildings must be managed.

The methodology was applied to the public real estate of the Campania region and especially to its historical, disused healthcare buildings. To this end, a database that collects all the public buildings located in the Campania region was created and potential indexes were computed. The main results of the analysis are reported in Section 4 (Discussion). It is to be stressed that transformability is mainly related to the type of load-bearing structure: masonry proves to be less transformable than an r.c. frame, while for buildings with the same structure, spatial and distribution issues and constraints concerning the external walls and roof are critical.

The added values proposed in this paper are both a conceptual procedure for assessing the potential and transformability of historical but disused monumental hospitals and a BIM-based application to facilitate such evaluations, assessing the potential and transformability of buildings. Further studies, based on parametric model-checking processes considering technological-functional requirements of intended uses (e.g., social housing, social services, etc.), are underway.

**Author Contributions:** Conceptualization, L.D., S.D. and G.A.; methodology, L.D., S.D. and G.A.; software, G.M.; validation, L.D. and G.M.; formal analysis, L.D. and S.D.; investigation, L.D. and S.D.; resources, L.D., S.D. and G.A.; data curation, L.D.; writing—original draft preparation, L.D., S.D. and G.M.; writing—review and editing, L.D. and G.A.; visualization, L.D., S.D. and G.M.; supervision, L.D. and G.A.; project administration, L.D. and G.A.; funding acquisition, L.D., S.D. and G.A. Abstract: G.A.; Section 1: L.D. and S.D.; Section 1.1: S.D.; Section 1.2: L.D.; Section 1.3: G.A. and G.M.; Section 2: G.A., L.D. and S.D.; Section 2.1: S.D.; Section 2.2: L.D.; Section 2.2.1: L.D.; Section 2.3: G.M.; Section 3.1: S.D.; Section 3.2: L.D.; Section 3.3: L.D.; Section 4.1: L.D.; Section 4.2: L.D. and G.A.; Section 5: L.D. and G.A. All authors have read and agreed to the published version of the manuscript.

**Funding:** This research was partially funded by the PON AIM project E61G18000550008, activity code AIM1849341-3.

**Institutional Review Board Statement:** Not applicable.

**Informed Consent Statement:** Not applicable.

**Data Availability Statement:** Data sharing is not applicable.

**Acknowledgments:** We thank Marcello Raiano, Daniela Volpe, Martina D'Alessio, and Federica Ambrosi De Magistris for their support with data acquisition.

**Conflicts of Interest:** The authors declare no conflict of interest.

## Appendix A

SCHEDA N. ______
DATA __/__/____
RILEVATORE

______________

### RILIEVO SPEDITIVO DEL PATRIMONIO AD USO PUBBLICO CAMPANO

**LOCALIZZAZIONE GEOGRAFICO-AMMINISTRATIVA** — **OGGETTO** — Sezione 1

Provincia ______________

Comune ______________

Località ______________

Indirizzo ______________

Nord/Lat. __________ Est/Long. __________

Denominazione______________

Epoca prima costruzione______________

Stato manutenzione generale ❑ottimo ❑ buono ❑ discreto ❑ scadente ❑ pessimo

Proprietà______________

**DESTINAZIONE D'USO ATTUALE** — Sezione 2

❑ museo ❑ archivio ❑ biblioteca ❑ struttura sanitaria ❑ non utilizzato ❑altro ______________

❑ culto ❑ uffici ❑ abitazione ❑ servizi ❑ struttura ricettiva-albergo

**ACCESSIBILITÀ ESTERNA / SERVIZI** — Sezione 3

Accessi pedonali n. ___ Accessi carrabili n. ___ Larghezza strada principale ____ Parcheggi nelle vicinanze: ❑Si ❑No

Trasporto pubblico in prossimità: ❑autobus ❑ metro ❑ treno ❑ tram Aree verdi nelle vicinanze: ❑Si❑ No

Servizi nelle vicinanze: ❑ scuole ❑poste/banche ❑market ❑ presidi sanitari ❑ farmacie ❑altro______________

**CARATTERISTICHE DEL SITO** — Sezione 4

Pianeggiante ❑ Centrale ❑ Centro storico ❑

In rilievo ❑ Semicentrale ❑ Zona residenziale ❑

A valle ❑ Periferica ❑ Area commerciale / industriale ❑

In prossimità della costa ❑ Suburbana / extraurbana ❑ Zona agricola ❑

**POSIZIONE** — Sezione 5

❑ Zona altamente urbanizzata

❑ Isolato ❑ Connesso con altri edifici ❑ Zona mediamente urbanizzata

**DATI MORFOLOGICI** — Sezione 6

| | Regolare | Non regolare |
|---|---|---|
| Pianta | ❑ | ❑ |
| Elevazione | ❑ | ❑ |
| Disposizione muri interni | ❑ | ❑ |
| Disposizione aperture | ❑ | ❑ |

| Forma in pianta | ❑ rettangolare | ❑ rett allungata | ❑ a L |
|---|---|---|---|
| | ❑ a C | ❑ a corti | ❑ altro |

❑porticati ❑logge ❑corti ❑ giardini ❑parcheggi interni

❑corpi annessi ❑ corpi aggettanti ❑ terrazzi

**DATI DIMENSIONALI** — Sezione 7

Sup. territoriale/lotto/p.lla:________ Sup. coperta:________ H gronda:______ H colmo:______ Vol. lordo:______

**DATI FUNZIONALI E TECNOLOGICI** — Sezione 8

Piani fuori terra n. ___ Piani entro terra n. ___ Vani scale n. ___ Ascensori n. ___

Struttura verticale: ❑ muratura ❑ telai c.a. ❑ misto Copertura/e: ❑ a falde ❑ piana

Orizzontamenti: ❑ putrelle in ferro e voltine ❑ orditura/e in legno ❑ latero-cementizio ❑altro______________

Presenza di lesioni negli elementi portanti verticali: ❑ Si ❑No Solai/coperture imbarcati/pericolanti: ❑ Si ❑ No

Infissi: ❑ legno ❑ metallo ❑PVC ❑vetro singolo ❑ vetro-camera Stato di conservazione:______________

Tompagnatura: ❑ tufo ❑calcare ❑laterizio ❑ forati ❑ altro__________ Spessore medio:________cm

**CONTESTO URBANISTICO-AMBIENTALE** Qualità/pregio... — Sezione 9

| ... architettonico: | ... ambientale: | ... del contesto architettonico: | ... del contesto ambientale: |
|---|---|---|---|
| ❑ ottimo ❑ buono | ❑ ottimo ❑ buono | ❑ ottimo ❑ buono | ❑ ottimo ❑ buono |
| ❑ discreto ❑ scadente | ❑ discreto ❑ scadente | ❑ discreto ❑ scadente | ❑ discreto ❑ scadente |
| ❑ pessimo | ❑ pessimo | ❑ pessimo | ❑ pessimo |

**Figure A1.** Rapid visual inspection form with nine sections: (1) "general data and geographical-administrative location"; (2) "current functional use"; (3) "external accessibility and services"; (4) "characteristics of the site"; (5) "position of the building"; (6) "morphological data"; (7) "dimensional data"; (8) "functional and technological data"; (9) "qualities and values".

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
