# Peer review of "Assessment of Disused Public Buildings: Strategies and Tools for Reuse of Healthcare Structures"

_sustainability, doi:10.3390/su14042361_

Round 1

Reviewer 1 Report

The paper presents a good topic related to. Assessment of disused public buildings: strategies and tools for reuse  The attached file contains many comments to improve the paper before publication.

Author Response

The authors thank the reviewer for the useful comments that deeply improved the general quality of the paper. Concerning the improvement of the linguistic quality, the paper has been revisited by a native English speaker and deeply modified according to his suggestions. Furthermore, the paper has been reduced in length (3000 word less) and simplified, especially in “Methodology” and “Discussion and Conclusion” section.

A point-by-point response to the reviewer's comments is provided in the attached file.

Best Regards,

PhD Lorenzo Diana
Corresponding Author
On behalf of the Authors

Reviewer 2 Report

The article focuses on a relevant topic and offer an interesting methodology to address the extended public buildings unused/abandoned. Congratulations to the authors.

The article, in my opinion, needs only few modifications/ corrections to be published, also if some further addition would improve it.

L.28-29 check punctuation.

L42. "Many" should be replaced for an estimated amount/porcentage.

L61. PNRR and PNC should be explained for non Italian readers.

L84. Also if stated later, do not focus only on economical aspects!

L332. "Given the purposes of the present paper"... seems that the score is assigned just for this publication.

L610. better: reuse ??

L 676 Fig. 8 8.c seems not to correspond with plans neither with  Fig. 9 model... in 8.a & 8.b would be interesting to clearly outline the unused section of the building.

L 682.  Figure 9 caption has to be modified.

Improvement suggestions: Given the importance of the automatization process through Revit plugin, it could be interesting for the scopes of replicability and understanding of the procedures, to add a section dealing with the programming aspects of the plugin, and also include more pictures, apart of fig. 11.

Bibliography suggestions: It could be worth to have a look at the following research:

https://www.sciencedirect.com/science/article/pii/S2095263519300792

Author Response

The authors thank the reviewer for the useful comments provided. The paper has been improved accordingly. A point-by-point response is sent in the attachments.

Best Regards,

PhD Lorenzo Diana
Corresponding Author
On behalf of the Authors

Reviewer 3 Report

This is research looks into developing a procedure to use second-hand building materials in new infrastructure development. I think the study is worthy of publication as this part of waste recycling is understudied, and the work is a valuable contribution to the theory and practice of built environment circular economy. I have the following suggestions to improve the quality of the work.

One line showing the research problem could be a good starter to the abstract

Long sentences in the abstract should be avoided

Sever editing is needed (this is a must)

No need to use phrases such as ‘the authors’ in the abstract

In the introduction, and perhaps across the text, please introduce and use concepts such as recycling, upcycling, circular economy. The following sources might be useful:

  • Shooshtarian, T. Maqsood, S. Caldera, T. 2022. Transformation towards a circular economy in the Australian construction and demolition waste management system. Sustainable Production and Consumption. 30 (2022), 89-106.
  • Shooshtarian, S., M.R Hosseini, T. Kocaturk., M. Ashraf and T. Arnel. 2021. The Circular Economy in the Australian Built Environment: The State of Play and A Research Agenda. Faculty of Science, Engineering and Built Environment, Deakin University, Australia.

Please also introduce some statistics on construction activities and the industry profile in Italy

Please check if some text of 1.2 and 1.3 can be tabulated

Figure 2 quality needs to be improved

Not sure if Figure 3 is needed to be inserted in the methodology section, please consult other reviewers’ opinions; if any, I feel that it needs to be added as an appendix

It might be outside of the study focus, but it is worth it if the authors also talk about the main barriers to using recycled materials in new buildings in the discussion; the following paper might be useful:

  • Shooshtarian, S Caldera, T Maqsood, T Ryley. 2020. Using recycled construction and demolition waste products: A review of stakeholders' perceptions, decisions, and Motivations. 5 (31), 1-17.

A lot of tables presented in the paper should be merged properly

The methodology section is way too long; please make it shorter

Figure 12 might not be needed

Please consider merging figures to reduce their number to 7 max

I suggest authors separate discussion and conclusions- it is an easy rearrangement

On page 26/31 there is a long paragraph that repeats the aim, objectives and results; while it is necessary to refresh readers mind about these, its length is unnecessarily extended.

Some minor inconsistencies in the reference list; please inspect them

Author Response

The authors thank the reviewer for the useful comments that improved the general quality of the paper. All the comments have been considered.

Best regards, 

PhD Lorenzo Diana,
Corresponding Author
On behalf of the Authors

Reviewer 4 Report

This study proposes an integrated method (HIBM + AHP) to evaluate potential strategies to recover disused healthcare structure. This method is applied to selected Italian case studies, while the reasons behind this selection are not clearly explained.  

A complete English revision by a native speaker is mandatory. The text is too long, many sections can be reduced and unnecessary  (or repeated) concepts should be removed. It very hard to follow the workflow and believe that this model can truly adopted in other contexts. I suggest to identify the contribution of this study and reduce the text to 8000 words.

Author Response

The Authors thank the reviewer for the useful comments that deeply improved the paper. The paper has been completely revisited. Many sections changed, reduced, or reorganized.

Some reasons have been given for the selection of the Italian Case studies (in the Abstract, in the Introduction, and in the Case Study sections). In addition, the authors tried to simplify the approach, to avoid repetitions, and statements considered unnecessary or not clear.

Concerning the English quality, the paper has been revisited by a Native English speaker and the text has been modified accordingly.

Furthermore, the revision of the paper brought to a reduction of its length: in the first submission, the total amount of pages was 31 (written 26), while in the current version, pages in total are 26 (written 21). The written part is now limited to 8,700 words, close to the goal of 8,000 asked by the reviewer, and some figures have been merged.

In the attachment, a point-by-point answer is provided.

Best regards

PhD Lorenzo Diana
Corresponding Author
On behalf of the Authors

Round 2

Reviewer 3 Report

Thanks for revising the paper; I am partly satisfied with the changes made. However, some of the changes made need careful attention as they are either not addressed or sufficient. I encourage authors to pay close attention to the nature of each comment and see how they can take advantage of the suggestions provided. My comments are as below:

  1. Table 1 is not clear; some more text could be more useful for the readership
  2. Figure 1 the tip of the pyramid is cut and should be shown
  3. I acknowledged the authors’ explanation as to why merging tables do not look good, but I still think that this will help make the paper concise
  4. I cannot see where the methodology is reduced- can the authors please specify the removed parts
  5. The work needs more careful editing – still, flaws can be seen here and there. This paper, after publication (provided that comments are addressed), will be seen by many many researchers due to its interesting and useful nature, so I encourage authors to spend a good time on editing the work to be useful for authorship around the world
  6. Except for one figure, I cannot see any other figures moved to appendices
  7. I don’t understand why authors have two sections named for discussion, 3.4. Discussion of results and 4. Discussion – are not they referring to results interpretation? If so, then all should be under 4. Discussion with some subheadings if needed.
  8. Please see the following references that do not comply with MDPI/Sustainability referencing style – there are heaps of other references in the list that needs careful attention. Furthermore, none of the journal names is abbreviated as advised by MDPI- I suggest that authors use MDPI created endnote file to convert their work to the preferred style.

  • Crova, C. LE LINEE GUIDA DI INDIRIZZO PER IL MIGLIORAMENTO DELL’EFFICIENZA ENERGETICA NEL 825 PATRIMONIO CULTURALE. ARCHITETTURA, CENTRI E NUCLEI STORICI ED URBANI: UN AGGIORNAMENTO 826 DELLA SCIENZA DEL RESTAURO. In Proceedings of the LE NUOVE FRONTIERE DEL RESTAURO. Trasferimenti, 827 contaminazioni, ibridazioni; 2017.
  • Buda, A.; Hansen, E.J. de P.; Rieser, A.; Giancola, E.; Pracchi, V.N.; Mauri, S.; Marincioni, V.; Gori, V.; Fouseki, K.; López, 832 C.S.P.; et al. Conservation-compatible retrofit solutions in historic buildings: An integrated approach. Sustainability 833 (Switzerland) 2021, 13, doi:10.3390/su13052927
  • Campisi, T.; Acampa, G.; Marino, G.; Tesoriere, G. Cycling Master Plans in Italy: The I-BIM Feasibility Tool for Cost and 840 Safety Assessments. Sustainability 2020, 12, Page 4723 2020, 12, 4723, doi:10.3390/SU12114723

Good luck with further changes.

Author Response

The authors thank the Reviewer for these further comments that helped the paper to be more effective. All the comments have been considered and improved the general quality of the paper.

In the attachment, a point-by-point response to reviewer's comments.

Best Regards,

the Corresponding Author,
On behalf of the Authors 

Reviewer 4 Report

The paper has been revised and can be published.

Author Response

The authors are pleasant to thank the reviewer for the useful work done for this review and especially for the previous one that substantially improved the quality of the paper. The authors tried to address all the comments and solve all the issues.

In the attached files, a point-by-point response to reviewer's comments.

Best regards

the Corresponding Author
On behalf of the Authors

Round 3

Reviewer 3 Report

Thanks for addressing the comments; again, these two comments need to be addressed carefully:

  1. Table 1 is not clear; some more text could be more useful for the readership

AUT: The authors thank the reviewer for the comment. A new caption for the table has been defined: “Criteria for heritage building reuse retrieved from the analyzed literature.” and further explanations are given in the text when the table is introduced: “In Table 1 a short summary of the literature comparison concerning evaluation criteria for heritage building reuse is depicted.”

I expected to see some changes in the table text, not the caption. Anyway, you can keep it this way. maybe the authors just (in the text) indicate different criteria listed in the previous literature would be a better explanation than a short summary of the literature

  1. I don’t understand why authors have two sections named for discussion, 3.4. Discussion of results and 4. Discussion – are not they referring to results interpretation? If so, then all should be under 4. Discussion with some subheadings if needed.

AUT: The authors thank the reviewer for the comment and for catching this typo. Section 3.4 is now named “Comments of results” while section 4 “Discussion”. One of the two directly provides results’ interpretation of the applied tool while the other underlines the meaning of the research. Still think the same way both of these can be presented under discussion 4.1. the application of tool and 4.2 research meaning (or other titles that are apt)

Author Response

The authors thank the reviewer for the comments that emerged in the three revisions provided. In consequence of the work of the reviewer, the general quality of the paper improved strongly. In attachment a point-by-point response.

Best Regards
